# A Unique Grubbing Head Prototype for Environmentally Friendly and Sustainable Stump Removal

Luboš Staněk, Ladislav Zvěřina, Radomír Ulrich and Eva Abramuszkinová Pavlíková *

Department of Engineering, Mendel University in Brno, 61300 Brno, Czech Republic
* Correspondence: eva.pavlikova@mendelu.cz

**Abstract:** Stumps left behind after the felling of trees represent an important source of renewable energy that could be used for fuel mixtures as a sustainable solution. The subject of this research was to determine the influence of tree species, stump diameter size, and subsoil on the time required for stump processing. Evaluated parameters included the mean time for one stump's processing; the stump processing time based on the stump diameter; different soil types and tree species; and the tree species type (coniferous, broadleaved). The research was conducted in the territory of the Czech Republic in 2020/2021. There were 287 stumps and 6 tree species in total. The stumps were uprooted using a new prototype of grubbing head, developed at Mendel University in Brno, attached as an adapter on the boom of a JCB JS 220 LC excavator with a tracked undercarriage and was controlled remotely from an excavator cab. Research results confirmed that the processing time of one stump depends on the stump diameter (GLM), and the time needed for the processing of one stump increased with an increase in stump diameter in all experimental sites. An equation was suggested to predict the time needed to work on one stump.

**Keywords:** forest operations; mechanization; sustainable management; stump removal; forest biomass; time efficiency; grubbing head





## 1. Introduction

To reduce dependence on imported fossil fuels, many experts predict that stump grubbing in forests is a sustainable solution [1] because forest biomass represents one of the most reliable and widespread renewable fuels and hence sources of energy. The interest in utilizing renewable resources to partially replace fossil fuels in generating energy is increasing in the European Union. In Finland, for example, similarly as in many other woody countries, forest biomass is considered a sustainable and readily available source of energy [2]. In these countries, a great part of the input materials for biofuels are logging residues (branches, tree tops, stumps) that are left in clear-cut areas after felling [3]. Wood fuels have been increasingly used in the last century. In Finland, the share of wood-based fuels, including forest biomass and industrial by-products, was 75% in 2017 [4]. In forest ecosystems, a great part of biomass is underground (roots, stumps) and can be a potential source of renewable energy. Therefore, it is expected that the use of biomass for energy is likely to be more intensive in the near future. This is also indicated by the Paris Climate Agreement [5] and the growing $CO^2$ content in the atmosphere [6,7].

Logging results in numerous clear-cut areas in formerly closed forests [8], and there is a lot of logging residue left behind in these clear-cut areas that could be used for energy purposes. Bark, stumps, roots, branches, tree tops, and assimilatory organs remaining in the forest after the main felling as logging waste without any direct use represent 35%–40% of produced dendromass [9]. Stumps are one source of forest biomass [10] and are obtained together with their roots by grubbing [11].

Stumps have been grubbed for energy purposes in Scandinavian countries since the 1970s [12], namely, those of Norway spruce (*Picea abies*) [10]. In the last ten years,

stump grubbing has become more topical in the forest practice of Nordic and Baltic countries [13–16]. Although stump grubbing for energy purposes is currently an uncommon practice, it can reduce the costs of preparing clear-cut areas for the subsequent regeneration of forest stands [17]. In Finland, for example, the grubbing of stumps and large roots is common [18] and has been gradually increasing since 2000. There was a great boom in 2006–2007 when the stump removal area increased by 50% [19]. In 2010, the size of stumped areas was approximately 20,000 ha in Finland [6], which was an increase of ca. 20% in the same year compared with the preceding year [20]. Stump grubbing reached a peak in this Nordic country in 2010–2013 with 1.1 mil. $m^3$ of grubbed stumps [4]. In this method, stumps and roots are grubbed, cut, and shaken so that earth and undesirable materials are removed. Then, they are piled next to the extraction line, where they are dried and cleaned by sun and rain [21]. Forwarders then transport them to an open storage place, where they stay for several months. From there, they are transported to end users in the form of whole stumps or chips [22].

The current situation, the so-called bark beetle disaster, offers a large amount of wood mass in the form of fuelwood. In the Czech Republic, the most damaged area is the northeastern part of the country. When grown-up trees infested by bark beetle are felled, extensive clear-cut areas are left behind with unused wood mass in the form of stumps and roots, which decay within a few years. However, stumps should not be left to gradual decay because wood-decaying fungi may occur in some localities (*Heterobasidion annosum* (Fr.) Bref.), which can also infest the root systems, and there is a risk that new plantations on regenerated plots can be infested by this fungal pathogen [23]. Using this wood mass for further processing is therefore desirable. One of the methods used to grub such wood waste is to employ grubbing heads.

Stump grubbing used to be common in the Czech Republic until recently, where it is now only used in some forms of forest management or deforestation. A method used most frequently in CR is grubbing stumps with a dozer blade. Stumps are piled into mounds or terrain depressions and left for natural decomposition. Stump "burial" is also locally used, where a trench is made by an excavator into which stumps are pushed and covered with earth [11]. A disadvantage of this method is that the wood mass cannot be further used.

One of the reasons why a new prototype of the grubbing head was developed was the widespread bark beetle calamity at the time, which offered consumers a large amount of wood material in the form of firewood. However, after the harvesting of these "bark" mature trees, extensive forest clearings were left, on which there was a large amount of unused wood in the form of stumps and roots, which would rot in a few years. Thus, the idea was to extract, process, and monetize this wood material in the form of wood chips for energy purposes. The new prototype of the grubbing head could therefore bring new economic opportunities for large-scale mass harvesting in all forests.

The advantage of the grubbing head is that it can harvest stumps in a gentle way with minimal disturbance of the soil. The machine on which it is attached can move along forest roads and does not necessarily have to enter the clearing. Since the machine does not enter clearings, there is also no compaction of the soil when driving and crossing the machine. Another advantage of the head is the ability to move the tree stump or its parts after it has been pulled out of the ground, until the soil and other unwanted materials are separated from the tree stump. The grubbing head can be used from machine-operating trails, and soil compaction by machine crossing can be avoided.

Another advantage is the lower part of the body of the head, where it is able to adjust, level, compact, and prepare the area after extracting the stump for subsequent mechanized forest restoration. The grubbing head used for tearing stumps is characterized by the fact that, when raised above the ground, the base of the head and the splitting arm are gradually spread apart and compressed, thereby releasing soil and stones from the lower spaces of the body of the head and between the root system. The soil falls into place after the stump, and the area is leveled in such a way that mechanized planting of the toll area is possible (or subsequent soil preparation, where necessary, directly with the lower part of the head

body). The head is controlled remotely between the body of the head and the splitting arm, which fit together. When the body of the head and the splitting arm are gripped over the stump, they split it and pull it out of the ground piece by piece.

For the above reasons, a new prototype of grubbing head was developed at the Faculty of Forestry and Wood Technology, The Department of Engineering, Mendel University in Brno, as a creative output from the project MPO no. FV 40031 "Multi-purpose modular system of grubbing stumps and other commodities". The project solution time is 5/2019–11/2022. The prototype, which served to gain the wood mass, was developed by company STS Prachatice, a.s. We estimated that stumps could be grubbed in the Czech Republic on an area of min 1000 ha, which would represent an annual source of about 50 thous. m$^3$ of stump wood with a stump wood stock of ca. 50 m$^3$/ha in the country [24].

## 2. Literature Review

### 2.1. Dendromass and Stump Removal

In forestry, dendromass for energy can be gained as fuel wood from logging or as logging waste left in the forest after felling in the form of branches, cut-offs, stumps, or roots. Stumps are not commonly used for energy in the Czech Republic in spite of the fact that stump removal was identified as long ago as the 1930s as one of the technologies in forestry with the greatest potential to ensure supplies of solid biofuel [25].

Stupavský et al. claim that logging residues are still a little-used source of renewable energy, with the exception of Scandinavian countries [26]. Melin et al. and Alam et al. are of the same opinion, claiming that in Finland, a great part of current forest harvesting for electric energy from dendromass is represented by logging residues [27,28], i.e., tree tops, branches, foliage, as well as stumps and skeleton roots. The grubbing of stumps and large roots in Scandinavia is also confirmed by Juntunen and Herrala-Ylinen [6] and Persson and Egnell [7], who inform that stumps, together with roots, are also grubbed in North America. In Latvia, forest dendromass is becoming even more important for forest owners and the forest industry, where logging residues are used for biofuels and the technology is widely accepted in both state and private forests [25]. According to von Hofsten, the main tree species in Scandinavia on which stump grubbing is focused is Norway spruce (*Picea abies)* [29]. Köstler et al. claim that thanks to their shallow root system, stumps of Norway spruce are great for stump removal, easy to lift, and damage to soil is not too extensive [16,30]. Von Hofsten states that the methods that were used for stump removal in 2006 involved the removal of the above-ground part of the stump and roots over 5 cm in diameter [29]. This resulted in harvestable stump biomass consisting of 32% of above-ground stump wood and 68% of root stump wood because stump wood nearly always includes also bark and fine roots [31].

As another technique for removing stumps from the clearing, for example, a dozer with a blade or cutters could be used, which grind and spread the wood material from the stump in its immediate vicinity. The disadvantage of these techniques is that the stumps cannot be used for energy purposes after mining.

The most similar to our prototype is the extraction of stumps with grubbing adapters, which are also on the booms of excavators or ripper teeth on a hydraulic excavator. However, they only pull out the stump, leave it in place, and do not repair the ground. The glade left after the extraction of stumps by other grubbing heads is not ready for full-scale mechanized afforestation, and another technique must be used for its preparation. None of the other adapters work with a system to pry the stump out of the ground; they pull the stumps up. They also require more digger strength and weight [11].

### 2.2. Stump Removal and Forest Management

The amount of stumps and large-diameter roots available on the clear-cut area after felling depends on the type of forest management, e.g., intensity and timing of the improvement in felling operations. These have an essential influence on the growth rate and further development of forest stands [32]. The process of one stump's processing includes several

work operations: stump uprooting, splitting it into more parts, shaking, cleaning, turning of a hydraulic arm, placement on the pile, and surface leveling. The result of our time study of these operations can provide information about time management that can be used to improve the quality of forest management.

As stumps consist of wood and bark of trees, they can be used to gain more renewable resources for biofuels from forest stands. This is why the felling of trees is sometimes followed by stump removal with the use of heavy machines [33], usually excavators with special teeth for the extraction of stumps, which can split the stumps into smaller pieces. Wood mass from stumps and roots represents 23%–25% of stem wood biomass both in spruce and pine [17]. Finér et al. estimated the total biomass of stump systems (without fine roots) in a 140-year-old stand of Norway spruce to be 21,875 kg.ha$^{-1}$ [34] and total stand biomass to be 101,943 kg.ha$^{-1}$. This represents stump systems comprising 21% of the total biomass of trees. Alam et al. inform that stumps and large roots in a mature forest stand represent ca. 25%–30% of the total tree biomass [28]. A year later, the same authors published a finding that stumps and roots could increase the total production of biomass (energy biomass and stem wood) by approximately 21%–36% [35]. Merilä et al. estimate that, together with large-diameter roots, stumps of Norway spruce make up 38% of the total tree biomass [36]. Hakkila reasons that in Norway spruce stumps, wood density grows from the stump to the roots because the growth in the vicinity of the stump is faster, and wood mass increment is thus higher [31]. Kalliokoski et al. state that tree anchoring is one of the most important functions of tree roots, and this is why most trees produce dense timber, particularly at the stem base, i.e., stump [37,38].

Uri et al. also claim that the volume of grubbed stumps may be notable and can even exceed 100 m$^3$.ha$^{-1}$. In Estonia, for example, the biomass of stumps harvested from Norway spruce in the regions of Myrtillus and Oxalistypy amounted to 44–55 t.ha$^{-1}$, and their energy content was 294 MWh.ha$^{-1}$ [16].

*2.3. Bioenergy and Stamping*

Stumps and their roots have also become an important source of bioenergy due to growing fears of climate change. In Sweden, it is estimated, for example, that stump removal could replace 2.5%–5% of the energy currently generated from fossil fuels [39]. Björheden is of a similar opinion and claims that the use of bioenergy from logging waste [40], which is usually left behind on the clear-cut area after the felling of trees, is increasing due to worries of climate change and increasing demand for bioenergy. Literature sources differ as to the energy content in stumps [25]. According to studies conducted in Finland [41], it is possible to gain ca. 140–160 MWh.ha$^{-1}$. The Tekes company cited 200 MWh.ha$^{-1}$ [42]. Von Hofsten is convinced that stump removal has the potential to produce 5–10 terawatthours (TWh) per year [29]. Neruda et al. point out that dendromass from forest management can be used for energy via direct incineration [11], i.e., without a modification of its dimensions or, in the case of wood, after cutting, splitting, chipping, or crushing. A similar rule for wood also applies to grubbed stumps. Other authors [11,43,44] stress the fact that within time management, there needs to be time for stumps to dry out before burning. Simanov claims that the calorific value of biomass decreases with an increase in moisture content therein and vice versa [43]. The author assumes this is due to the fact that heat from incineration is consumed during the evaporation of water contained in the biomass. He states that the calorific value is a very important property of dendromass determined for energy use as well as related moisture content. Pastorek et al. claim that forest dendromass always contains a min of 10% water [44], and the moisture content in freshly cut timber is 40%–60%, and this is why the timber has to be left to dry out for at least a year, by which its moisture content will be reduced to 15%–30%, which is more appropriate for incineration. Neruda et al. add that in burning wood from standing trees, the share of energy needed to dry it out is higher than in other fuels. Since the range of wood moisture content is large, the range of its effective calorific value is also large [11]. The situation is the same in burning stumps or roots, and it is recommended, therefore, to

let this organic material dry out before burning, e.g., in incinerators, to a moisture content of a min of 30%.

### 2.4. Advantages of Stump Removal

Although the main purpose of stump removal is wood biomass, the method also has some other advantages. Stump grubbing disturbs a considerable part of the upper soil layer, which results in soil loosening and mixing. Since a part of the soil surface is mineralized after the stump removal, the need for site preparation for the subsequent forest regeneration is reduced along with the costs [45], which represents unexpected costs for the forest owners. In addition, the initial nitrogen availability for new targeted species is improved thanks to the increased mineralization [46,47]. This is how natural regeneration can be supported by extracting stumps [45] and how stump removal can be considered a method of site preparation for forest regeneration. At the same time, seedlings of undesirable species will be eliminated together with their root systems. Persson and Egnell add that stump removal has no negative influence on the growth of the next forest generation [7]. This is confirmed by Egnell and Hyvönen et al., who point out that their studies did not record any long-term adverse environmental effects of stump removal on the undergrowth vegetation and stand productivity [48,49].

Petersson et al. and Rahman et al. claim in agreement that an undisputable advantage of stump extraction is lower damage to seedlings caused by the pest *Hylobius abietis* [50,51], which improves the survival of planting stock because the female pests lay eggs in the stumps [52].

Cleary et al. are certain that one of the advantages of stump removal is also reduced damage to the forest by fungal pathogens [23]. Stumps left in clear-cut areas are the main point of infection for pathogens to enter a new generation and remain the dominant source of infection [53], as fungus can survive in stumps for up to 46 years [54]. For example, transmission by *Heterobasidion* sp. is known to occur in the regeneration of coniferous stands [54]. The issues related to stump extraction have been discussed in some previous studies [23,55]. Woodward et al. and Garbelotto and Gonthier claim that the rot of stumps and roots caused by *Heterobasidion annosum* (Fr.) Bref. is the most devastating disease of conifers in the northern hemisphere, namely, in North America, Europe, China, Japan [56–58], and the south of Russia. This suggests that it is a significant problem of forest regeneration in several areas of the world, especially where intensive forest management is applied [58], as the number of root contacts increases during intensive periods of tree growth, which often supports the transition of fungal pathogens from infected stump roots to healthy roots of replanted trees [59]. Woodward et al. estimated that damages caused in Europe by *Heterobasidion* sp. are equal to 790 mil. EUR per year [56]. Vasaitis et al. found out that the removal of stumps can reduce the distribution and infection of molds on root systems in the soil, which usually affects spruce stands of subsequent generations [60].

### 2.5. Disadvantages of Stump Removal

Stump removal can have negatives, too. According to Walmsley and Godbold and Hellsten et al. [61,62], it removes nutrients from the site (namely through the removal of small roots) and results in soil compaction. Zemánek and Neruda explain that the movement of forest machines across the stand causes interaction between the machine undercarriage and the soil surface by means of the contact area of wheels or tracks [63]. According to Eklöf et al. and Kiikkilä et al. [64,65], other disadvantages of stump grubbing consist in soil disturbance, which may cause the leaching of nutrients and heavy metals from the clear-cut area, and risk of erosion. The risk of soil disturbance is also confirmed by Berg et al. [66], who inform that the average size of disturbed soil is 6 m$^2$ per harvested stump, and that this size increases exponentially with the growing size of the stump. Berg et al. fear a possible reduction in the amount of dead wood [67], which might potentially reduce important fungi, mosses, and insects on the site.

## 3. Materials and Methods

The main goal of this research was to investigate the performance of a newly developed device, the unique grubbing head prototype, for environmentally friendly and sustainable stump removal using a time study. This was conducted in different locations, examining the effect of stump diameter and various tree species and their type (coniferous/broadleaved) on time consumption. The sub-goal was to establish a mean processing time and to determine time intensity and efficiency in the production of forest biomass. Production efficiency in this contribution was evaluated on the basis of the time that is needed to process one stump.

Authors are aware of the fact that the price of the stump biomass is related to more factors than processing time only. It can be influenced by stump weight, amount of energy it contains, etc.

The data used for this study were collected from November 2020 to July 2021. Three experimental sites (clear-cut areas) were chosen in the Czech Republic, which differed in natural, climatic, and soil conditions so that the results of measurements could be sufficiently unbiased. The localization of selected research sites is shown in Figure 1.

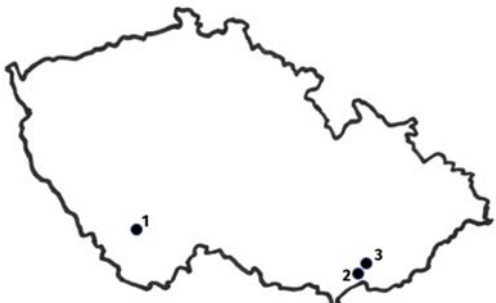

**Figure 1.** Localization of selected research site on the territory of the Czech Republic. The first experimental site was situated in the South Bohemian Region, the Prachatice district (Boubín forest enterprise Boubín, forest district of Netolice–1). The remaining two sites were in the South Moravian Region, Hodonín district (Management plan area (LHC) Strážnice, district of Ratíškovice–2; LHC Strážnice, district of Bzenec, Pískovna–3).

Detailed descriptions of selected sites are presented below. In 2020–2021, forest stands covering the sites were felled. Stumps were extracted three up to six months after the felling of trees. The activity of stump removal was performed on each site by a different operator. All operators had several years of experience with the given activity, and their performance was comparable for the purpose of comparing the research results.

Prior to the time study, all stumps to be measured on the clear-cut areas were numbered, the species was determined, and the stump diameter was measured using the forest caliper. The measurement of stump height was taken from 10 to 30 cm from the ground, based on stump diameter. The larger the diameter, the higher the stump height.

Stump diameter was determined as an average of the longest and shortest sides. The obtained data were then recorded in the prepared tables. Time data about the duration of individual operations were recorded on the numbered stumps using an electronic stopwatch for accurate measurement of time intervals. In each cycle of stump grubbing, which consists of respective partial operations, one stump was processed. There were 287 stumps and 6 tree species in total recorded at the research sites.

The time of processing one stump is a work process that consists of several operations: uprooting, splitting, cleaning from earth and surface treatment, and/or soil surface compaction at minimum damage. The whole process was video recorded, which was the basis for time study and analysis of specific activities. Results were compared in relation to subsoil type, size of stump diameter, and tree type or species. The prototype of the grubbing head (ca. 2 tons—see below) was attached as an adapter to the tracked excavator JCB JS 220 LC (Table 1). This type of excavator was used because some former research

activities confirmed the hypothesis that optimal performance requires an excavator of higher performance class and higher weight; otherwise, the performance of the grubbing head would not reach its potential.

**Table 1.** Technical parameters of JCB JS 220 LC [68].

| | |
|---|---|
| Weight | 23,002 t |
| Transport length | 9.622 m |
| Transport width | 2.77 m |
| Transport height | 3.266 m |
| Width of tracks | 600 mm |
| Boom | MB |
| Maximum reach | 9.361 m |
| Depth range | 6.096 m |
| Digging force | 126.6 kN |
| Engine manufacturer | JCB |
| Engine type | 448 DIESELMAX |
| Engine power | 129 kW |

### 3.1. Grubbing Head Characteristics

Figure 2 shows a grubbing head (1) in the technical solution for the attachment to the excavator boom, which contains a support arm (2) whose bottom side ends in two protruding ripping tips (3). In the upper part of the support arm (2), there is a hollow cover (9) of the head (1). In the cover (9) front, there are holes (11) for the placement of a splitting arm (4), and at the back, there are holes (15) for the placement of a cylinder (7) controlling the splitting arm movement. The splitting arm (4) is placed in the holes (11) in a swinging arrangement. It is curved, ending in a spike (24), and directed against and between the ripping tips (3) on both sides of the support arm (2). On the opposite end of the spike (24), the splitting arm (4) is provided with an anchoring device (8) for a cylinder (7) piston rod (6), which is mounted and swings on the respective axis. The splitting arm (4) swings by extending and retracting the cylinder (7) piston rod (6). The splitting arm (4) is provided with a cover (12) in the upper part, which acts as a lock. On the other side, the movement is limited by a stop (14) on the support arm (2). On the bottom wall of the support arm (2) of the grubbing head (1), there is a base (5) that serves for the compaction and treatment of soil from which the stump was removed for the successful regeneration of clear-cut areas.

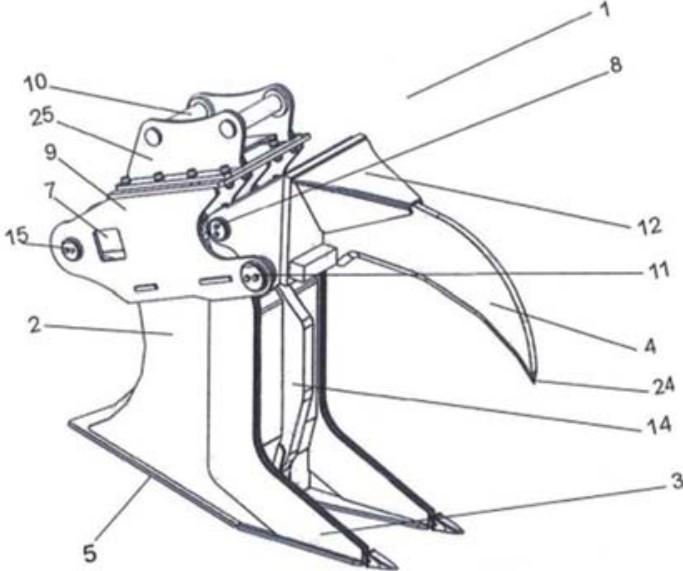

**Figure 2.** Grubbing head.

Figure 3 shows that on the upper side of the cover (9), there is a fixture (25) with two axes (10). Fixtures (25) serve to couple the grubbing head (1) with the boom (16) of the excavator, which is not shown. The fixture (25) is provided with a flange (13) with two axes, on one of which the boom (16) is revolving, and on the other (21), there is an arm (20) with one end bent, whose swinging is controlled in the bend (22) by the piston rod (18) of the boom (16), with its other end revolving directly on the boom (16) in the anchoring device (23). The second end of the piston rod (18) is in the cylinder (17) of the excavator boom (16). Other parts were explained in Figure 2.

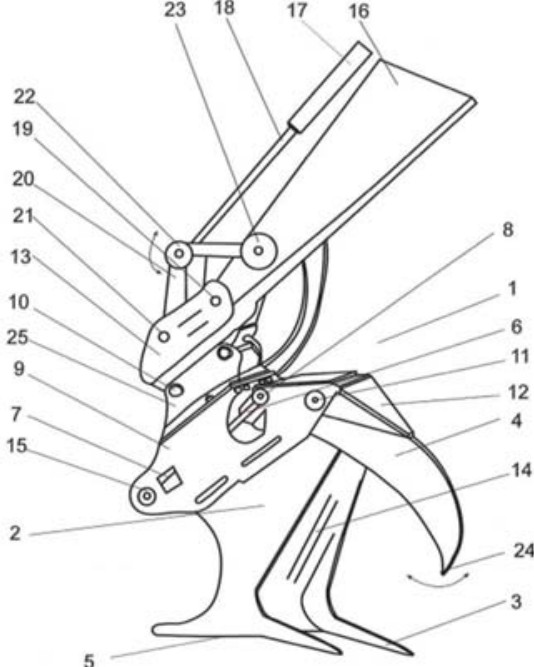

**Figure 3.** Grubbing head attachment.

The processing time of one stump is affected by a number of factors. As a part of the total time consumption of the production chain, it is required to reduce it as much as possible. However, it consists of several operations—phases to be conducted during one stump processing. Each of these operations is necessary to achieve higher performance, better product quality, and higher production efficiency in accordance with environmental protection. During the stump processing, some operations may be repeated more than once—depending on the stump diameter or the size of the root system—and these individual operations are the following:

- Stump separation—the process whereby a stump or its parts is torn out from the ground and lifted up. If the stump or root system is larger, it must be cleaved first in the soil with the help of a cleaving arm of the grubbing head and then pulled out of the ground in parts to a height of about one meter. The advantage of this grubbing head is the possibility to move the stump, or its parts, after being pulled out of the subsoil until the soil and other undesirable materials are separated from the pulled stumps. Therefore, in this subsequent position, the grubbing head must be shaken to release the unwanted soil and stones that were located on the cut stump or between the root system. This operation can be repeated until the complete removal of mined stump dendromass.
- Boom rotation—once the stumps or their parts are free of impurities, they are held by the grubbing head attached to the excavator´s boom and moved to a selected place to be stored in piles. The number of piles and the distance between them are mainly determined by the reach of the excavator´s boom, which carries the grubbing head.

- Storage—on the selected storage place, cleaned stumps or their parts are released from the grubbing head to be stacked appropriately. Extracted dendromass is then left to dry out until the required moisture value.
- Back rotation of the boom—after the storage operation, the excavator´s boom returns to the stump extraction site, and the first and next operations are followed and repeated until the whole stump dendromass is extracted
- Surface treatment—the used technology may not disrupt the site environment, and the soil surface disturbance must be particularly avoided in order to preserve the forest ecosystem and prepare conditions for site reforestation. During some of the procedures mentioned above, the soil falls either into the space where the stump is torn out or all around the site. For that reason, the surface is aligned with the movements of the excavator´s boom, and the bottom part of the grubbing head is eventually adjusted and compacted. This operation cannot be performed when using other types of grubbing heads.

The head was held on the excavator boom in its fixture, where the interlocking of the head body and the splitting arm was controlled remotely. Thus, the stump was split and pulled out from the ground by parts to a height of ca. one meter. In the following head position above the ground, the head base and the splitting arm were gradually spread and pressed together, by which earth and stone on the cut stump or in its root system were shaken off. When uprooted, the stump or its parts were moved until all undesirable materials were separated from the stump. The soil fell into the space after the uprooted stump, and the soil surface was leveled and compacted by means of the lower part of the head body. Then, the stumps or their parts were placed on piles using the boom.

Five hypotheses were used to reach the aim of this study:

**H1.** *There is a linear relationship between the time of one stump's processing and the stump diameter.*

**H2.** *The mean time of one coniferous stump's processing will be lower than the mean time of one broadleaved stump's processing.*

**H3.** *The most demanding time for one stump's processing will be for pedunculated oak stumps.*

**H4.** *The stump-separation stage is the most time-consuming operation in the processing time of one stump.*

**H5.** *The effect of tree, location, and diameter, or the interaction of the effect, is statistically insignificant.*

*3.2. Methodology of Time Study*

1. The evaluation and comparison of the mean time is based on groups established according to stump diameter size on different experimental sites. The comparison of the mean times for processing one stump at the selected experimental sites for each group is according to the stump's diameter size;
2. The evaluation and comparison of the mean time of coniferous and broadleaved stumps;
3. The comparison of mean times for processing one stump is according to tree species occurring in all selected experimental sites with no regard to the subsoil type and soil characteristics;
4. The determination of the percentage share of the time of individual work operations, the sum of which is the total time to process one stump;
5. The application of the general linear model with all predictors to find out the effect of tree, location, and diameter on time. IBM SPSS Statistics software was used.

Time performance and its evaluation were focused on the total time to process one stump. The total time for processing one stump does not take into account traveling from one stump to another. One stump's processing included the following work operations: stump uprooting; its splitting into more parts; shaking—cleaning (separation of soil from

stumps); turning of the hydraulic arm, placement on the pile; and surface leveling—treatment (compaction). In stumps with large dimensions, the work operations had to be repeated several times.

### 3.3. Characterization of Research Sites

#### 3.3.1. Site 1: Forest Enterprise LZ Boubín, Forest District of Netolice

The site, according to forest management plans and regional plans of forest development, is classified as a fresh Beech stand (*Fagetum mesotrophicum*). The soil in this locality is freshly moist and loamy. The soil type is Cambisol. Terrain relief on the research site is sloping. The tree species represented on the site was exclusively the Norway spruce (*Picea abies*), and altogether 157 stumps were assessed, with a diameter ranging from 15 to 77 cm.

In order to evaluate and compare the mean time of processing one stump, the stumps were classified into three reference groups by their diameter. The first group contained all stumps with a diameter $\leq 30$ cm, the second group included stumps with a diameter ranging from 31 to 60 cm, and the third group was stumps with a diameter of 61 cm or more. The stump numbers of the respective groups are presented in Table 2.

**Table 2.** Numbers of stumps in diameter groups on the site of LZ Boubín, Netolice.

| Tree Species | Number of Stumps in Diameter Groups (Pieces) | | | Total |
| --- | --- | --- | --- | --- |
| | ≤30 cm | 31–60 cm | >60 cm | |
| Norway spruce | 46 | 97 | 14 | 157 |

#### 3.3.2. Site 2: LHC Strážnice, District of Ratíškovice

The site is classified as an enriched hornbeam–oak stand (*Carpineto-Quercetum acerosum deluvium*). Soil is loamy, drought-prone, humic, and weakly gleyic. Soil types are oligotrophic to mesotrophic Brunic Arenosols. Terrain relief on the experimental site is flat.

There were 5 tree species and 110 stumps recorded on the site. Stump diameters ranged from 11 to 140 cm: pedunculate oak (*Quercus robur*), 20 stumps of diameters from 21 to 140 cm; Norway spruce (*Picea abies*), 37 stumps of diameters from 24 to 66 cm; field maple (*Acer campestre*), 14 stumps of diameters from 11 to 82 cm; robinia (*Robinia pseudoacacia*), 17 stumps of diameters from 15 to 60 cm; and small-leaved linden (*Tilia cordata*), 22 stumps of diameters from 11 to 75 cm. Regarding the fact that the stumps occurring on the site exhibited greater diameters, four reference groups according to stump diameter size were defined to compare values measured on all sites. The species representation and the number of stumps in each group on the site are presented in Table 3.

**Table 3.** Species representation and stumps by diameters—LHC Strážnice, district of Ratíškovice.

| Tree Species | Stumps by Diameter (pcs) | | | | Total |
| --- | --- | --- | --- | --- | --- |
| | ≤30 cm | 31–60 cm | 61–90 cm | >90 cm | |
| Norway spruce<br>Pedunculate oak<br>Field maple<br>Robinia<br>Small-leaved linden | 30 | 61 | 11 | 8 | 110 |

#### 3.3.3. Site 3: LHC Strážnice, District of Bzenec, Pískovna

The site is sandy with specific drift sands and flat terrain relief. There was only one tree species on the site—Scots pine (*Pinus sylvestris*)—and 20 stumps with diameters ranging from 14 to 65 cm.

3.3.4. Representation of Stumps on Sites

The representation and number of stumps in each group are shown in Table 4. As for the comparison of (coniferous/broadleaved) tree species, in the LHC Strážnice, district of Ratíškovice, it was Norway spruce (*Picea abies*) with 37 stumps. On the site of LZ Boubín, district of Netolice, the number of stumps of Norway spruce amounted to 157, and their diameters ranged from 15 to 77 cm. Thus, altogether, there were 194 Norway spruce (*Picea abies*) stumps. Another assessed species was Scots pine (*Pinus sylvestris*), with 20 stumps whose diameter ranged from 14 to 65 cm. The total number of assessed stumps of coniferous tree species was 214.

**Table 4.** Representation of tree species and stumps (including their diameter) for the comparison of time required for processing one stump.

| Tree Species | Stumps (pcs) | Stump Diameter Range (cm) | Ø Time (mm:ss) |
|---|---|---|---|
| Norway spruce | 194 | 15–77 | 02:48 |
| Scots pine | 20 | 14–65 | 02:37 |
| Robinia | 17 | 15–60 | 01:56 |
| Field maple | 14 | 11–82 | 02:12 |
| Small-leaved linden | 22 | 11–75 | 02:10 |
| Pedunculate oak | 20 | 21–140 | 05:18 |
| Total | 287 | | |

The number of stumps recorded in broadleaved tree species was 73, with the shares of individual species being as follows: pedunculate oak (*Quercus robur*), 20 stumps of diameters ranging from 21 to 140 cm; field maple (*Acer campestre*), 14 stumps of diameters ranging from 11 to 82 cm; robinia (*Robinia pseudoacacia*), 17 stumps of diameters ranging from 15 to 60 cm; small-leaved linden (*Tilia cordata*), 22 stumps of diameters ranging from 11 to 75 cm.

**4. Results**

All data and results from previous measurements on all sites were used for comparison. The data analysis was completed using statistical calculations outputs, including graphs showing maximum and minimum times of stump processing, Q25, Q75 quartiles, median and mean time for processing one stump in units of seconds, which is a sum of the values of quantitative statistical sign divided by the set extent. For a good informative value of measured results, arithmetic means were used in their evaluation and comparison.

**Hypothesis 1 (H1).** *There is a linear relationship between the time of one stump's processing and the stump diameter.*

Mean times for processing one stump were compared only on the basis of tree species criterion (Table 4). For Hypothesis 1, the soil type and diameter of individual stumps were excluded.

*4.1. Site 1: Forest Enterprise LZ Boubín, District of Netolice, with Cambisol Soil Type*

Results of our research (Figure 4) indicate that the mean time for processing one stump in the group of stumps with a diameter of up to 30 cm was 99 s, i.e., 1 min and 39 s. In the group of stumps with diameters ranging from 31 to 60 cm, it was 204 s, i.e., 3 min and 24 s. The last group on the given site with stump diameters larger than 61 cm exhibited a mean time for processing one stump of 280 s, i.e., 4 min and 40 s.

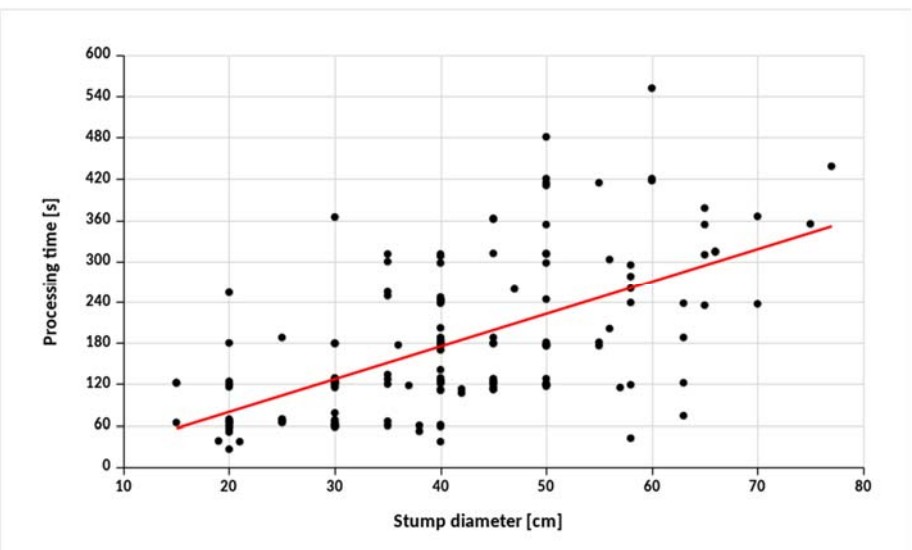

**Figure 4.** Statistical evaluation of time for processing one stump on the site of LZ Boubín, district of Netolice.

The results clearly show that the time for processing one stump depends on the size of its diameter and increases with an increase in diameter.

### 4.2. Site 2: LHC Strážnice, District of Ratíškovice, with Brunic Arenosols Soil Type

The measured values of our research (Figure 5) indicated that the mean time for processing one stump in the group of stumps with a diameter of up to 30 cm was 78 s, i.e., 1 min and 18 s. The mean time for processing one stump in the group of stumps with a diameter from 31 to 60 cm was 139 s, i.e., 2 min and 19 s. The mean time for processing one stump in the third group of stumps with a diameter from 61 to 90 cm was 251 s, i.e., 4 min and 11 s. The mean time for processing one stump in the last group containing stumps with a diameter over 90 cm was 471 s (7 min and 51 s). The results show that the time for processing one stump depends on the stump's diameter. Similarly, as in the previous site, the dependence between the stump diameter and the mean time of processing one stump is linear.

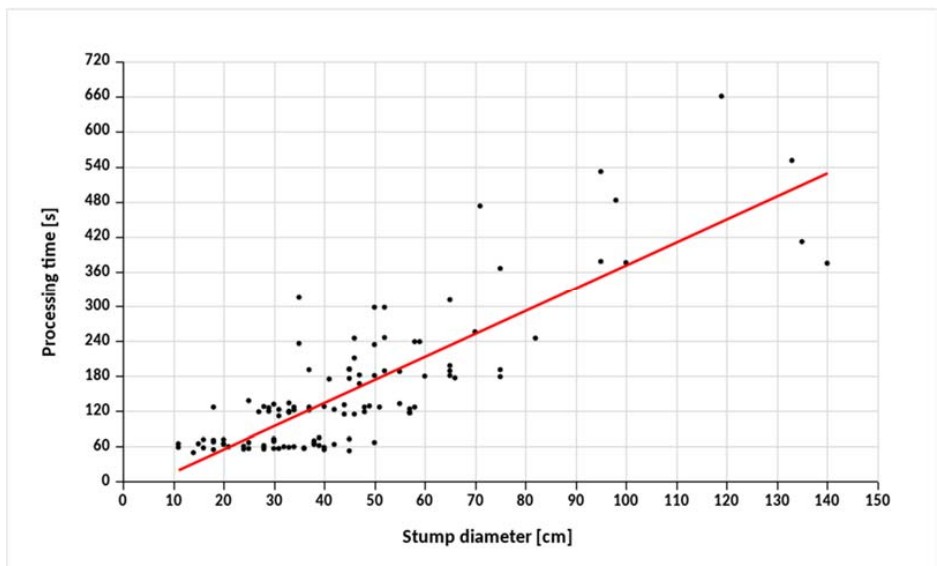

**Figure 5.** Statistical evaluation of time for processing one stump on the site of LHC Strážnice, district of Ratíškovice.

*4.3. Site 3: LHC Strážnice, District of Bzenec, Pískovna, with Sandy Soil Type*

The recorded values (Figure 6) showed that the mean time for processing one stump in the group with stump diameters of up to 30 cm was 77 s, i.e., 1 min and 17 s. In the group of stumps with diameters from 31 to 60 cm, the mean time for processing one stump was 173 s, i.e., 2 min and 53 s. In the last group of stumps with diameters from 61 to 90 cm, the mean time for processing one stump was 209 s, i.e., 3 min and 29 s.

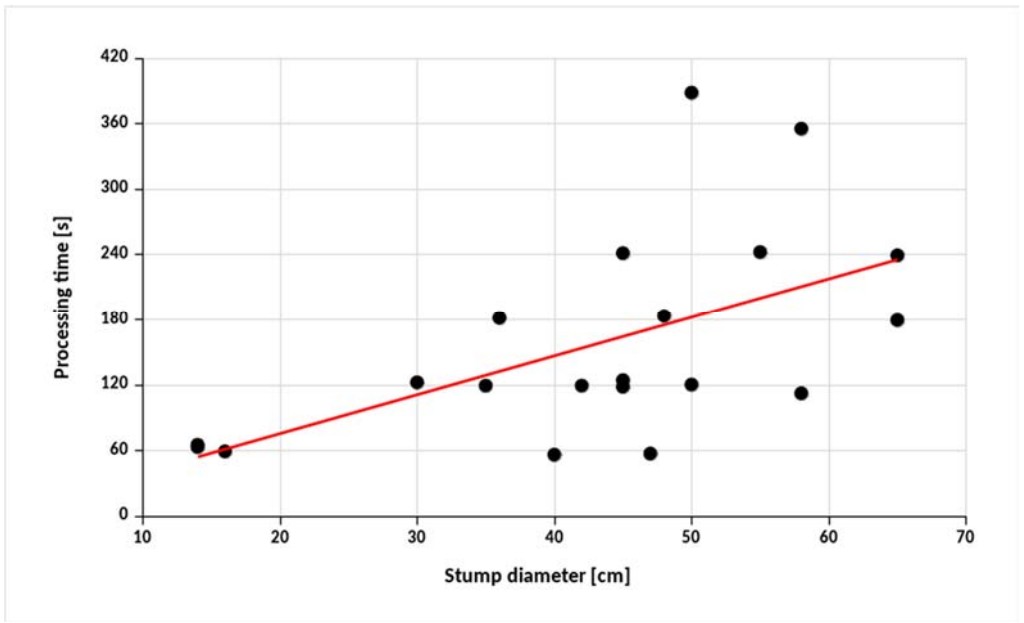

**Figure 6.** Statistical evaluation of time for processing one stump on the site of LHC Strážnice, district of Bzenes, Pískovna.

The results show that the time for processing one stump depends on the stump diameter. The time required for processing one stump increases with the increase in stump diameter.

Values presented in Table 5 indicate that the shortest time for processing was recorded in stumps from the group of diameters smaller than 30 cm. Stumps with the largest diameters needed the longest time for processing in all experimental sites with different soil types.

**Table 5.** Comparison of time for processing one stump at the respective sites by stump diameter.

| Site | Mean Time for Processing One Stump (mm:ss) | | | |
|---|---|---|---|---|
| | <=30 cm | 31–60 cm | 61–90 cm | >90 cm |
| LZ Boubín, district of Netolice | 01:39 | 03:24 | 04:40 | - |
| LHC Strážnice, district of Ratíškovice | 01:18 | 02:19 | 04:11 | 07:51 |
| LHC Strážnice, district of Bzenec, Pískovna | 01:17 | 02:53 | 03:29 | - |

Hypothesis 1 was confirmed. The relation between the processing time of one stump and diameter is linear. This was observed in all three sites with different soils.

**Hypothesis 2 (H2).** *The mean time of one coniferous stump's processing will be lower than the mean time of one broadleaved stump's processing.*

The soil type and diameters of individual stumps were not included. Figure 7 presents a comparison of the mean time for processing one stump for coniferous and broadleaved tree species. In conifers, the mean time for processing one stump was 167 s (2 min 47 s). In broadleaved tree species, the mean time for processing one stump was 179 s (2 min 59 s). The hypothesis was confirmed, but the difference in our sample was not high. Based on our field results, in terms of the stump processing speed, it is not important whether the tree species is coniferous or broadleaved.

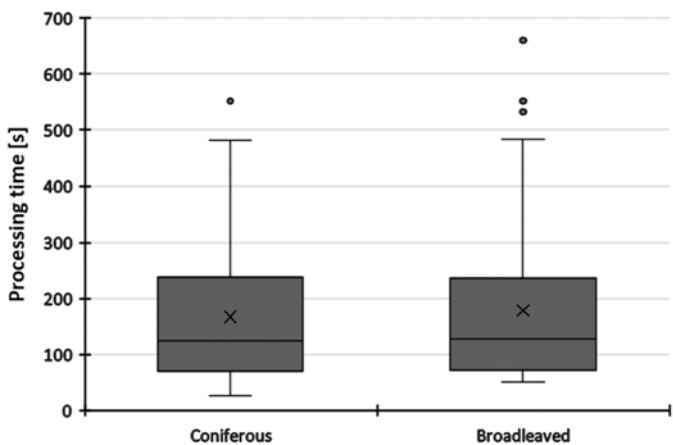

**Figure 7.** Comparison of time for processing one stump by tree species type (coniferous, broadleaved).

**Hypothesis 3 (H3).** *The most demanding time for one stump's processing will be for pedunculated oak stumps.*

Figure 8 shows that a considerable difference in the mean time for processing one stump compared with the other tree species can be seen in pedunculate oak (*Quercus robur*), which needs the longest time, 318 s (5 min 18 s). The lowest time—116 s (1 min 56 s)—was required for processing one robinia (*Robinia pseudoacacia*) stump. In Norway spruce (*Picea abies*), the mean time for processing one stump was 168 s (2 min 48 s). In Scots pine (*Pinus sylvestris*), it was 157 s (2 min 37 s); in field maple (*Acer campestre*), 132 s (2 min 12 s); and in small-leaved linden (*Tilia cordata*), the mean time for processing one stump was 130 s (2 min 10 s) (also see Table 4 for results). Thus, Hypothesis 3 was confirmed.

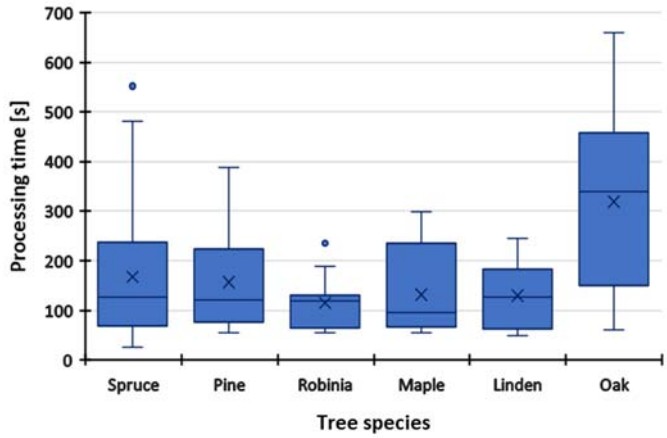

**Figure 8.** Comparison of time for processing one stump by tree species.

**Hypothesis 4 (H4).** *The stump-separation stage is the most time-consuming operation in the processing time of one stump.*

Percentage shares of individual work operations when processing one stump were determined based on the data collected during the time study of work operations for the grubbing of one stump.

After processing all data based on several field measurements, the average processing time of one stump, including the treatment of the subsoil after extraction and moving the excavator to the next stump, was found to be between two and three minutes. The total time of 131 s is the average time, including all working operations, but without transport to the next stump. If transportation is also included, the time reaches about three minutes. Detailed percentage distribution of individual work operations spent on processing one stump is shown in Table 6. The greatest amount of time is used for separation and storage, followed by the surface treatment operation. This was valid in all three case studies.

**Table 6.** Processing time of work operations.

| | Separation | Boom Rotation | Storage | Back Rotation | Surface Treatment | Total |
|---|---|---|---|---|---|---|
| Total time (mm:ss) | 38:48 | 05:38 | 35:03 | 07:55 | 26:13 | 1:53:35 |
| Ø time on stump (s) | 45 | 7 | 40 | 9 | 30 | 131 |
| Ø time (%) | 34 | 5 | 31 | 7 | 23 | 100 |

Hypothesis 4, stating that the stump-separation stage is the most time-consuming operation in the processing time of one stump, was confirmed. It should be added that almost the same amount of time is needed for the storage of stumps.

**Hypothesis 5 (H5).** *The effect of tree, location, and diameter, or the interaction of the effect, is statistically insignificant.*

With the General Linear Model (GLM), we tested the following hypothesis: The effect, or the interaction of the effect, is statistically insignificant. The dependent variable was time, and various effects and their interaction were tested, such as type of tree, location, and diameter.

The GLM with all predictors was significant only for the diameter factor, as shown in Table 7. The other effects, such as tree, place, and their interactions with diameter, are insignificant.

**Table 7.** Estimates and tests of effect size (tree, place, diameter, place/diameter, tree/diameter, tree/place).

**Dependent Variable: Time**

| Source | Type III Sum of Squares | df | Mean Square | F | Sig. | Partial Eta Squared |
|---|---|---|---|---|---|---|
| Corrected Model | 1,938,309.681 [a] | 13 | 149,100.745 | | | |
| Intercept | 2524.589 | 1 | 2524.589 | 23.583 | 0.000 | 0.529 |
| Tree | 2673.071 | 4 | 668.268 | 0.399 | 0.528 | 0.001 |
| Place | 1082.366 | 1 | 1082.366 | 0.106 | 0.980 | 0.002 |
| Diameter | 317,967.949 | 1 | 317,967.949 | 0.171 | 0.679 | 0.001 |
| Place * Diameter | 16,077.141 | 1 | 16,077.141 | 50.293 | 0.000 | 0.156 |
| Tree * Diameter | 16,680.925 | 4 | 4170.231 | 2.543 | 0.112 | 0.009 |
| Tree * Place | 0.000 | 0 | . | 0.660 | 0.621 | 0.010 |
| Error | 1,726,000.193 | 273 | 6322.345 | . | . | 0.000 |
| Total | 11,932,790.000 | 287 | | | | |
| Corrected Total | 3,664,309.875 | 286 | | | | |

[a] R Squared = 0.529 (Adjusted R Squared = 0.507). Tree: 1—pine, N = 20; 2—oak, N = 20; 3—maple, N = 14; 4—locust, N = 17; 5—linden, N = 22; 6—spruce, N = 194. Place: 0/N = 157; 1/N = 20; 2/N = 110. * means combination of place/diameter/tree.

The problem with the model is the fact that data for all trees in all locations are not available. For this reason, we use another model with a place effect only for spruces and a model with only a tree effect without place.

As shown in Table 8, the General Linear Model was used only for spruces, which were the only ones found in two localities, 0 and 2, which were Netolice and Ratíškovice districts. From the table, it can be seen that the only predictor is the diameter. The interaction of place and diameter is also at the border of significance, which indicates a different influence of diameter for the two localities. Location 2, which is taken as a reference, has a regression value of 2.59, which means that with each centimeter increase in the diameter, the time increases by an average of 2.59 s. For location 0, this coefficient increases by 2.18, which means that with each centimeter increase in diameter, the time increases by an average of 4.77 s (Table 9).

**Table 8.** SPRUCES—estimates and tests of effect size (place, diameter, place/diameter). Locality—0: Netolice, 1: Pískovna, 2: Ratíškovice.

**Dependent Variable: Time**

| Source | Type III Sum of Squares | df | Mean Square | F | Sig. | Partial Eta Squared |
|---|---|---|---|---|---|---|
| Corrected Model | 816,053.546 [a] | 3 | | | | |
| Intercept | 54.090 | 1 | 272,017.849 | 38.106 | 0.000 | 0.376 |
| Place | 1082.366 | 1 | 54.090 | 0.008 | 0.931 | 0.000 |
| Diameter | 182,757.036 | 1 | 1082.366 | 0.152 | 0.697 | 0.001 |
| Place * Diameter | 16,077.141 | 1 | 182,757.036 | 25.601 | 0.000 | 0.119 |
| Error | 1,356,320.975 | 190 | 16,077.141 | 2.252 | 0.135 | 0.012 |
| Total | 7,630,037.000 | 194 | 7138.531 | | | |
| Corrected Total | 2,172,374.521 | 193 | | | | |

[a] R Squared = 0.376 (Adjusted R Squared = 0.366). * means combination of place and diameter.

**Table 9.** SPRUCES—parameter estimates.

**Dependent Variable: Time**

| Parameter | B | Std. Error | t | Sig. | 95% Confidence Interval Lower Bound | 95% Confidence Interval Upper Bound | Partial Eta Squared |
|---|---|---|---|---|---|---|---|
| Intercept | 9.362 | 58.192 | 0.161 | 0.872 | −105.423 | 124.148 | 0.000 |
| [Place = 0] | −24.116 | 61.933 | −0.389 | 0.697 | −146.281 | 98.049 | 0.001 |
| [Place = 2] | 0 [a] | . | . | . | . | . | . |
| Diameter | 2.585 | 1.367 | 1.891 | 0.060 | −0.111 | 5.280 | 0.018 |
| [Place = 0] * Diameter | 2.180 | 1.452 | 1.501 | 0.135 | −0.685 | 5.045 | 0.012 |
| [Place = 2] * Diameter | 0 [a] | . | . | . | . | . | . |

[a] This parameter is set to zero because it is redundant. * means combination of place and diameter.

As shown in Tables 10 and 11, the GLM with tree and diameter effects and their interaction in all data again show that only the diameter effect is significant. In the next table (estimation parameter), however, in addition to the variable diameter, there is also a significant parameter for tree = 5 (linden), and at the 10% significance level, the parameter for tree = 4 (locust), both interacting with diameter.

This means that the regression coefficient of 4.48 for diameter means that with each centimeter, the time increases, on average, by 4.48 s for the reference variable spruce; the others do not differ significantly, only for linden, the coefficient is lower by 1.89, or for locust, by 2.38.

Since the tree effect is insignificant, it is possible to switch to the simplest model, where only the diameter will be the explanatory variable, as shown in Tables 12 and 13.

**Table 10.** Estimates and tests of effect size (tree, diameter, tree/diameter).

| Dependent Variable: Time | | | | | | |
|---|---|---|---|---|---|---|
| Source | Type III Sum of Squares | df | Mean Square | F | Sig. | Partial Eta Squared |
| Corrected Model | 1,792,252.545 [a] | 11 | | | | |
| Intercept | 5772.734 | 1 | 162,932.050 | 23.934 | 0.000 | 0.489 |
| Tree | 16,801.436 | 5 | 5772.734 | 0.848 | 0.358 | 0.003 |
| Diameter | 438,464.709 | 1 | 3360.287 | 0.494 | 0.781 | 0.009 |
| Tree * Diameter | 38,004.256 | 5 | 438,464.709 | 64.409 | 0.000 | 0.190 |
| Error | 1,872,057.330 | 275 | 7600.851 | 1.117 | 0.352 | 0.020 |
| Total | 11,932,790.000 | 287 | 6807.481 | | | |
| Corrected Total | 3,664,309.875 | 286 | | | | |

[a] R Squared = 0.489 (Adjusted R Squared = 0.469). Tree: 1—pine, N = 20, 2—oak, N = 20, 3—maple, N = 14, 4—locust, N = 17, 5—linden, N = 22, 6—spruce, N = 194. * means combination of diameter and tree.

**Table 11.** Parameter estimates.

| Dependent Variable: Time | | | | | | | |
|---|---|---|---|---|---|---|---|
| | | | | | 95% Confidence Interval | | |
| Parameter | B | Std. Error | t | Sig. | Lower Bound | Upper Bound | Partial Eta Squared |
| Intercept | −15.881 | 19.432 | −0.817 | 0.415 | −54.136 | 22.374 | 0.002 |
| [Tree = 1] | 20.578 | 59.626 | 0.345 | 0.730 | −96.803 | 137.958 | 0.000 |
| [Tree = 2] | 53.169 | 45.629 | 1.165 | 0.245 | −36.658 | 142.997 | 0.005 |
| [Tree = 3] | 24.198 | 51.554 | 0.469 | 0.639 | −77.293 | 125.688 | 0.001 |
| [Tree = 4] | 57.911 | 55.611 | 1.041 | 0.299 | −51.565 | 167.388 | 0.004 |
| [Tree = 5] | 40.586 | 46.799 | 0.867 | 0.387 | −51.543 | 132.715 | 0.003 |
| [Tree = 6] | 0 [a] | . | . | . | . | . | . |
| Diameter | 4.484 | 0.452 | 9.921 | 0.000 | 3.595 | 5.374 | 0.264 |
| [Tree = 1] * Diameter | −0.932 | 1.321 | −0.705 | 0.481 | −3.533 | 1.669 | 0.002 |
| [Tree = 2] * Diameter | −0.725 | 0.670 | −1.083 | 0.280 | −2.044 | 0.594 | 0.004 |
| [Tree = 3] * Diameter | −1.131 | 1.233 | −0.918 | 0.360 | −3.558 | 1.296 | 0.003 |
| [Tree = 4] * Diameter | −2.378 | 1.449 | −1.641 | 0.102 | −5.230 | 0.475 | 0.010 |
| [Tree = 5] * Diameter | −1.893 | 1.058 | −1.790 | 0.075 | −3.975 | 0.189 | 0.012 |
| [Tree = 6] * Diameter | 0 [a] | . | . | . | . | . | . |

[a] This parameter is set to zero because it is redundant. * means combination of diameter and tree.

**Table 12.** Estimates and tests of effect size.

| Dependent Variable: Time | | | | | | |
|---|---|---|---|---|---|---|
| Source | Type III Sum of Squares | df | Mean Square | F | Sig. | Partial Eta Squared |
| Corrected Model | 1,701,983.493 [a] | 1 | | | | |
| Intercept | 495.786 | 1 | 1,701,983.493 | 247.189 | 0.000 | 0.464 |
| Diameter | 1,701,983.493 | 1 | 495.786 | 0.072 | 0.789 | 0.000 |
| Error | 1,962,326.382 | 285 | 1,701,983.493 | 247.189 | 0.000 | 0.464 |
| Total | 11,932,790.000 | 287 | 6885.356 | | | |
| Corrected Total | 3,664,309.875 | 286 | | | | |

[a] R Squared = 0.464 (Adjusted R Squared = 0.463).

**Table 13.** Parameter estimates.

| Dependent Variable: Time | | | | | | | |
|---|---|---|---|---|---|---|---|
| | | | | | 95% Confidence Interval | | |
| Parameter | B | Std. Error | t | Sig. | Lower Bound | Upper Bound | Partial Eta Squared |
| Intercept | −3.231 | 12.042 | −0.268 | 0.789 | −26.935 | 20.472 | 0.000 |
| Diameter | 4.037 | 0.257 | 15.722 | 0.000 | 3.531 | 4.542 | 0.464 |

Using this equation, we can predict the explained variable time.

$$Time = -3.23 + 4.04 * Diameter$$

## 5. Discussion

The main goal of this research was to investigate the performance of the unique grubbing head prototype for environmentally friendly and sustainable stump removal using a time study. Five hypotheses were used to reach this aim. The first hypothesis confirmed that there is a linear relationship between the time of one stump's processing and stump diameter. This was confirmed at all three research sites. The same was confirmed in GLM (Hypothesis 5), where predictors and the effect of tree and location on time were insignificant, with the only exception being the diameter factor.

The specific times in our time study provide information on how much average time is needed for coniferous or broadleaved stumps, what the difference is between tree species in time operations, and how much time is needed for specific operations with this grubbing head. Based on our results, we have suggested an equation that can predict how much time is needed to work on one stump.

Although the stump removal was performed on each site by a different operator, this had no important influence on the mean time of stump processing, as all operators have several years of experience with the given activity, and their performance is comparable. In general, it can be stated that the performance of any machine controlled by man depends on human performance, namely, on the machine operator's experience. The operators working with the machine equipped with the grubbing head had very similar long-term experiences, and differences in their performance were unrecordable, while the weight and performance class of the excavator is a basic prerequisite for the full use of the operational potential of the new prototype. Stump harvesting does not depend on machine operators as in felling or thinning, where the emphasis on the product and its final quality is more important. We are aware of the fact that the ideal situation would be work completed by the same operator. For practical reasons, this was not possible, so we selected operators with the same work experience. Although there might be some slight effect of the differences between operations, it does not influence the result of our findings.

A new prototype of a grubbing head (company STS Prachatice, a.s., Prachatice, Czech Republic) was used in this study as an adapter to the tracked excavator Model JCB JS 220 LC (company J. C. Bamford, Rocester, UK). There were 287 stumps processed during our research belonging to 6 tree species on 3 sites with different soil types. Research results showed that the time for processing one stump depends on the stump diameter. A similar opinion was published by Palander et al. [69], who confirmed, based on the results of their research, that the time for uprooting larger-diameter stumps was longer. Laitila et al. [70] used an excavator weighing 17 tons with a fork-type implement for lifting the stumps and claimed that the excavator required less time for processing stumps of up to 47 cm in diameter than for processing stumps with diameters over 47 cm. The same authors inform that the time needed for processing a stump of 35 cm in diameter is 51 s. Fredriksson [71] and Kärhä [72] add that the grubbing of large stumps requires heavier machines with greater stability and power (the same opinion as ours based on previous research studies). At the same time, they present an example from Finland where the

machines most frequently used for stump removal are those weighing 21 tons. Another advantage of large excavators is that the reach of their boom is greater, which increases their productivity, as they are capable of processing more stumps from one position without having to travel.

Our research also demonstrated that the time required for processing one stump increases with the increase in the size of the stump, in our case, with no regard to the soil type. We are aware of the limitations of this research related to the quality of the soil. For further research, we would suggest a more detailed analysis of soil composition.

Palander et al. [69] also claim that the relationship between time productivity and stump diameter is linear. Similarly, Laitila et al. [70] and Kärhä [73] found that changes in the time productivity of stump removal were different in diverse stands, and the variability grew in larger stump diameters. Palander et al. [69] assume that the time differences are likely to follow from different methods of stump removal, i.e., machines used and the influence of the person grubbing the stumps.

The unique grubbing head prototype for environmentally friendly and sustainable stump removal was introduced in this study in different locations, examining the effect of time, stump diameter, and various tree species. The authors believe that further research in this area can contribute to detailed knowledge about new and effective methods to process forest biomass for energy use. Further research could involve a focus on the detailed quality of soil composition or the human factor, meaning the quality of operators and their influence on productivity. Another topic could be the economic background and effectiveness of this work, which was not a focus of our study.

## 6. Conclusions

Operational tests of the grubbing head prototype contributed to new findings for using another source of energy, dendromass, that can be gained from logging waste. The design of the grubbing head prototype enables the processing of the stumps without apparent dirt, i.e., in quality comparable with other parts of dendromass, and at the same time avoids soil degradation and prepares the site for forest regeneration. As for future applicability, the results of this study can provide a basis for the evaluation or comparison of the time and economic effectiveness of using stump-removal technology. For this purpose, a sufficient set of uprooted and cleaned stumps was used, which were left in the clear-cut area after trees were felled on the selected sites. The localities were selected to ensure the difference in the types of subsoil to evaluate and compare the head performance not only according to the soil type but also according to the size of the stump's diameter and tree species.

The time performance of the grubbing head is significantly affected by the diameter of the extracted stump. This was confirmed on each of the three selected sites when the mean time for processing one stump increased with the increase in diameter size. The longer time for processing a stump with a larger diameter can be explained by the larger the stump diameter, the greater the amount of stump dendromass, including the root system, that has to be processed. Work operations such as splitting, turning, or placement can sometimes be repeated several times. However, a surprising fact was that soil types have almost no influence on the mean time for processing one stump, based on our field sites. It was demonstrated using a comparison that the groups of stump diameters at all three sites had approximately the same processing times for stumps, with differences of less than 10%.

The longest time for processing one stump was recorded for pedunculate oak (*Quercus robur*). This was not so surprising because stump diameters larger than 90 cm were represented in 45% of the total number of stumps in this species. An important factor can also be the shape and structure of its root system, which reaches depths considerably greater than, for example, the root system of Norway spruce (*Picea abies*), which is flat and shallow. As for the comparison of time for processing one stump according to the tree species, there were almost no differences among the results of other species.

The same situation was also recorded when comparing the mean time for processing one stump between coniferous and broadleaved tree species when the results were nearly

identical. The representation of tree species and stump diameters is not uniform, and this is why the comparison according to selected factors may appear biased or insufficient. Nevertheless, all research activities were performed according to local natural conditions and the scheduling possibilities of all participants, and the amount of collected data can be considered sufficient for using the research results in practice.

Data from all field measurements were processed and indicated that the mean time for processing one stump, including the treatment of the disturbed soil surface, was from two to three minutes. This is a time during which the process could not be feasible using current machines and technologies. Thus, an experienced operator is capable of processing, on average, about 20 stumps within an hour if the excavator does not have to travel to the next stump, depending on the density of the stumps.

Research on this topic is continuous, and the presented results are to be considered partial only. In the future, more data will be gathered from further localities that will differ in soil conditions. A comparison is also expected of this prototype with similar types of adapters already used in practice [11], both in respect of the purpose of their use and in terms of time and economic effectiveness.

The results of this study can be used by practitioners for better planning the time required for stump removal in consideration of different stump diameters.

**Author Contributions:** Conceptualization, L.S., L.Z., R.U. and E.A.P.; methodology, L.S., L.Z.; investigation, L.S., L.Z. and R.U; resources, L.S.; data curation, L.Z., E.A.P.; writing—original draft preparation, L.S., L.Z.; writing—review and editing, E.A.P.; visualization, L.Z., E.A.P. All authors have read and agreed to the published version of the manuscript.

**Funding:** This research was funded by the CR Ministry of Industry and Trade (MPO), grant number FV 40031, TRIO 4, project "Multi-purpose modular system of grubbing stumps and other commodities", project timeline: 5/2019–11/2022.

**Data Availability Statement:** Not applicable.

**Conflicts of Interest:** The authors declare no conflict of interest.

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
