# Peer review of "A Unique Grubbing Head Prototype for Environmentally Friendly and Sustainable Stump Removal"

_forests, doi:10.3390/f13091515_

Round 1

Reviewer 1 Report

Stumping is a terminology sometimes miswriting in the manuscript. 

I want to read some information about the forest, such as age, mean annual increment, or site production.

Also, I encourage the authors to add the stump weight and wood density for readers unfamiliar with temperate trees. This info will add a forest appeal and attract more readers looking for the biomass potential of stumps for energy.  This way, even the discussion can have an item about energy density.

Author Response

Thank you for your comments which improved the quality of our paper. We made major changes in the text and also included new statistical investigations (GLM).

1. We have changed "stumping" term into stump removal. It explains better the content.

2. You were interested in the information about the forest (age, site production, mean annual increment). This information is very difficult to get in a short time. This information is not publically available, it would need personal visit on sites which are in three different places in Czech Republic, get access to specific documents etc... For the purpose of our study it was not very important.

3. Similarly, the stump weight and wood density was not measured during our research. It is true that it could attract more readers. We will consider this for our futher research. For the objective which we had, this was not needed.  

Thank you for your time. 

Reviewer 2 Report

Dear authors,

I admire that you have developed the new device. I would like you to explain more clearly in the article why you have developed this and what you see as the advantages over other devices. I also think that the data you have collected can be better analysed. You can find detailed comments from me in the attached document.

Kind regards,

Reviewer

Author Response

Thank you for your comments which were important for increasing the quality of our paper. We tried to explain more clearly why we have developed this prototype including advantages over other devices. We made new analysis of data. 

Based on your comments, we have reformulated the objective, explained the motivation for new device and mentioned why it is environmentally friendly. We believe that our hypotheses bring a new knowledge about working with this prototype and we also addded hypothesis 5 for GLM. 

We applied changes as you suggested in Structure. Parts of the text were moved, reformulated or changed. We added information which you required in content part. More detailed statistical evaluation was done, which you find in tables 7-13. The analysis was done on total sample of stumps. 

We have changed the discussion and added sentence about usability of results.

As for the presentation of results, we have done only some changes such as: changed the form of graphs or added better explanation in the text. We think that the information provided is important for a reader, or easier (such as map).

Comments on individual passages: 

- we have changed the term stumping to stump removal

-original line 19 - yes - GLM model showed the result

-original lines from 45 to 396 we changed or taken into account

-original lines 436, 461 - results of GLM have details

-original lines 463 to 683 we changed or taken into account

-original line 742 - the literature reference number 24 was in the original article, so we did not change anything. It was probably misunderstanding.

Thank you very much for cooperation.

Reviewer 3 Report

The work is potentially very interesting for the scientific community, presenting the stumping operation in Czech Republic. 

Unfortunately, overall, the manuscript is poorly written and partially poorly organized, and there are a few major issues.

Author Response

Thank you very much for your comments. We improved the quality of our text by many changes in all chapters. It was important to add general linear model and hypothesis 5. The contribution can be interesting not only for experts in the Czech Republic, but worldwide.

Thank you again for your time. 

Reviewer 4 Report

Althoguht the introduction provides a suffient background, it should be improved in some statements. However, the paper lacks of scientific soundness. The hypotheses as well as data analysis are very simple. Discussion is inexistent. 

Author Response

Thank you very much for your comments. We improved the quality of our text by major changes in all chapters. It was important to add general linear model and hypothesis 5, followed by new tables from 7-13 with Estimates and Tests of Effect Size on Time. Major changes in text have been done.

Thank you for cooperation. 

Round 2

Reviewer 3 Report

Thank you for your efforts.

Unfortunately, overall, the manuscript is poorly written and partially poorly organized, and there are a few major issues.

Author Response

Thank you for your comments which were important for increasing the quality of our paper.

We tried to explain more clearly why we have developed this prototype including advantages over other devices. We have given hopefully better background in the introduction part, we have improved research design and also presentation of results. 

Specifically, we have introduced new hypothesis 5 and made a new data analysis using IMB SPSS software for GLM as can be seen from tables 7-13. The analysis was done on total sample of stumps.

We have reformulated the objective and explained the motivation for new device. We applied many changes as you suggested . Parts of the text were moved, reformulated or changed - introduction, literature review, contextualization of the topic, materials and methods. We have changed the discussion and added sentence about usability of results.

We have changed the term stumping to stump removal. 

Thank you very much for your cooperation.

Reviewer 4 Report

The manuscript has been sufficiently improved to warrant publication in Forests.

Author Response

Thank you for your time and your comment:

The manuscript has been sufficiently improved to warrant publication in Forests.